# Adenoviral Vectors for Gene Therapy of Hereditary Diseases

**DOI:** 10.3390/biology13121052

**Published:** 2024-12-16

**Authors:** Anna Muravyeva, Svetlana Smirnikhina

**Affiliations:** Laboratory of Genome Editing, Research Centre for Medical Genetics, Moskvorechye, 1, 115522 Moscow, Russia

**Keywords:** adenoviral vector, gene therapy, hereditary diseases, cystic fibrosis, ornithine transcarbamylase deficiency

## Abstract

Adenoviral vectors (AdVs) are engineered viruses used to deliver therapeutic genes to specific cells, offering promising solutions for the treatment of genetic diseases. However, their use is limited by issues such as strong immune responses and transient transgene expression. These limitations make them unsuitable for treatments that require long-term gene expression, such as some inherited diseases. This review examines clinical trials using AdVs for gene therapy of cystic fibrosis and ornithine transcarbamylase deficiency, their successes in preclinical testing and failures in practice, and discusses the underlying reasons for the failure of clinical trials. Understanding the reasons may help overcome these barriers to advances in gene therapy for inherited diseases. The review also highlights the achievements in overcoming these barriers. Scientists are modifying the outer structure of these vectors to more precisely target specific cells, attempting to reduce immune responses to AdVs, and improving gene delivery in cystic fibrosis by removing physical barriers such as thick mucus in the lungs. While these vectors are currently most useful for short-term applications such as vaccines and genome editing, ongoing research may open new doors for their use in more complex treatments. These advances have the potential to improve the effectiveness of gene therapy and offer hope to people living with incurable diseases.

## 1. Introduction

Adenoviruses are viruses measuring 70–100 nm in diameter, which include the icosahedral capsid along with all its components, including the fiber and knob structures, and contain a linear double-stranded DNA genome of 25–45 kb, depending on the type (Figure 1). The DNA includes inverted terminal repeats (ITRs) of approximately 100 bp at both the left and right ends of the genome, which contain replication origins. The viral DNA packaging signal (ψ), about 150 bp long, is located immediately adjacent to the left ITR. The genome also encodes early (*E1*–*E4*) and late (*L1*–*L5*) genes, which are expressed before and after DNA replication, respectively.

Each early gene region contributes uniquely to the virus lifecycle and its potential therapeutic applications. The *E1* region, particularly *E1A*, encodes proteins that induce mitogenic activity in host cells, pushing them into the S-phase of the cell cycle, a requirement for viral DNA replication. Additionally, E1A proteins stimulate the expression of other viral genes, initiating the transcriptional cascade required for efficient replication [1]. The deletion of *E1* renders adenoviruses replication-deficient, a key safety measure for their use in gene therapy. The *E2* region is responsible for viral DNA replication through its encoded proteins, including the viral DNA polymerase, the pre-terminal protein (pTP), and the DNA-binding protein. These proteins mediate the replication of the viral genome and are essential for the successful completion of the viral infection cycle [2]. The *E3* region encodes proteins that modulate host immune responses by downregulating major histocompatibility complex (MHC) class I molecules and inhibiting apoptosis. These mechanisms enable the virus to evade immune detection and maintain infection. Although vital for immune evasion during natural infection, the *E3* region is dispensable in vitro, and its deletion increases the transgene capacity of adenoviral vectors [3]. The *E4* region encodes proteins that influence host cell signaling and enhance the processing, transport, and translation of viral mRNA. Deletion of *E4* genes reduces immune activation and increases the stability of transgene expression, making advanced adenoviral vectors more efficient and less immunogenic [4].

The viral genome codes for about 40 proteins responsible for maintaining the adenovirus infection cycle, replicating viral DNA, packaging DNA, and assembling virions, as well as structural capsid proteins [5]. Currently, more than 116 types of human adenoviruses have been identified, as classified by the Adenovirus Working Group [6]. This extensive diversity highlights the ability of adenoviruses to infect a wide variety of tissues and hosts, with approximately 80% of the population harboring antibodies to one or more types. This makes adenoviruses valuable tools for gene therapy and vaccine development [7]. Adenoviruses are not exclusive to humans; they are also found in other species, which expands their potential for therapeutic applications. For instance, the chimpanzee-derived adenovirus vector ChAdOx1 has been effectively utilized in vaccine development, including the widely known Oxford–AstraZeneca COVID-19 vaccine [8].

The adenovirus infectious cycle is shown in Figure 2. Infection is initiated by the formation of a high-affinity complex between the virus’s globular knob domain and various membrane receptors, including CAR, CD46, and integrins, on the host cell surface [9]. Adenovirus penetration into cells occurs via endocytosis into the cytoplasm. Transcription of early and late genes occurs before and after viral DNA replication, respectively [10]. At the end of the infectious cycle (~24 h), viral proteins are synthesized in the cytoplasm. Adenovirus assembly occurs in the nucleus, and then virions exit into the cytoplasm with the destruction of the nuclear membrane, subsequently leaving the cells by their lysis. Additionally, adenovirus capsids often do not contain DNA, as significantly more structural proteins are produced than necessary for virion formation [5].

Adenoviruses have been isolated from many species, and in humans, they infect the respiratory and gastrointestinal tracts, causing mild respiratory or gastrointestinal diseases [11]. An adenovirus vector is a recombinant virus in which specific regions of its genome, such as the E1 or E3 region, are removed to make the virus replication-deficient or to increase its capacity to carry a transgene. While the E1 region is typically replaced with a transgene, the E3 region is often deleted without being replaced [12]. AdVs are constructed based on mastadenoviruses (infecting mammals)—members of the *Adenoviridae* family [13]. The prototype is AdV type 5, which naturally infects a large population of people and uses coxsackie and adenovirus receptors (CAR) to enter the cell [14]. It is important to note that different adenovirus types use different cell receptors for entry, which can be used to target AdVs to specific cell types, leading to higher transduction efficiency of these cells [15]. Researchers actively use AdVs for gene therapy of infectious diseases, cancer, and vaccine development [16].

## 2. Adenoviral Vectors for Gene Therapy

In this section, we summarize the key advantages and disadvantages of adenoviral vectors in gene therapy applications, as illustrated in Figure 3. This figure provides a visual overview of the strengths and limitations discussed below, offering a clear comparison of their potential uses in various therapeutic contexts.

AdVs have several advantages that make them potential candidates for the successful treatment of various diseases and vaccine development. Firstly, AdVs have high transduction efficiency, allowing them to deliver genetic material to target cells with high precision and efficiency, including both dividing and non-dividing cells, due to their broad tropism, which is ensured by the ubiquitous CAR expression on various cell types that the adenovirus vector uses for attachment and penetration into cells. In 1997, it was shown that CAR is the primary receptor for AdVs [17]. Additionally, AdVs provide high levels of transgene expression in cells, increasing the effectiveness of the therapeutic agent with lower doses of the vector introduced, thereby reducing the risk of side effects and increasing treatment safety. In nature, wild-type AdVs do not integrate into the host genome but remain inside the cell as an episome, providing temporary transgene expression [18]. These properties play an important role in the therapy of infectious and oncological diseases, as well as in vaccines, where temporary antigen expression is sufficient to stimulate the immune system and create protective immunity against the pathogen. Moreover, this feature makes AdVs attractive for genome editing technology. Thus, AdVs can deliver components necessary for editing, directly initiating the genome editing process and then quickly being eliminated from cells, reducing the likelihood of off-target editing—non-specific and unintended genetic mutations—outside the editing site. It is noteworthy that AdVs have a high capacity and can accommodate relatively large transgenes, making them suitable for delivering and expressing large genes or multiple genes simultaneously. Different generations of vectors can accommodate approximately 5 to 37 kb [19]. Finally, AdVs can be produced with high titers, facilitating their production and use in research and clinical practice, including vaccine production.

However, adenoviral vectors have drawbacks that may limit their use in gene therapy. AdVs have high immunogenicity and can induce cytotoxic T-cell or humoral responses with the formation of neutralizing antibodies. These immune reactions can reduce the vector’s effectiveness by preventing successful gene delivery. Additionally, adenovirus capsid components can enter the MHC class I processing pathway and cause cytotoxic responses in target cells [20]. These factors make adenoviral vectors among the most immunogenic viral vectors, posing an obstacle to their use in gene therapy. Moreover, AdVs lack the specificity due to their ability to infect many cell types, as cells in various organs have the CAR receptor. This has been confirmed by numerous in vivo studies on models of rabbits, pigs, dogs, mice, and rats [21,22,23,24]. Furthermore, due to insufficient specificity, high doses of AdVs are often administered to achieve effective transduction levels of target cells. High doses can cause inflammatory reactions such as increased cytokine levels, leading to serious side effects, especially with systemic administration of the vector [25]. Despite temporary transgene expression being advantageous for AdVs in cancer therapy, vaccine action, or genome editing methods, this same property is a disadvantage for using AdVs in gene therapy of hereditary diseases, specifically gene replacement therapy, which involves replacing a defective gene with a functional one. For this approach, prolonged and stable transgene expression is required. Finally, most people already have an immune response to adenoviruses due to previous exposures, such as past infections or vaccinations. Thus, antibodies to AdVs after the initial administration can reduce the effectiveness of repeated dosing [26]. 

In the context of gene therapy, AdVs are often compared with adeno-associated virus (AAV) vectors, as AAVs are among the most widely used platforms for treating hereditary diseases. Both vector systems have unique advantages and limitations, making them suitable for different therapeutic applications. AdVs have a much larger packaging capacity, accommodating transgenes of 5 to 37 kb, compared to the 4.7 kb limit of AAV vectors [27]. However, AdVs are highly immunogenic, provoking robust immune responses, whereas AAV vectors are relatively non-immunogenic, making them more suitable for long-term gene therapy [28]. AAVs also exhibit broader and more selective tropism due to the diversity of their serotypes, each targeting specific cellular receptors [29]. In contrast, AdVs primarily utilize the CAR receptor, which is widely expressed across many cell types, contributing to their broader but less specific tropism. These differences influence their therapeutic applications. AAV vectors are commonly used in gene therapy for hereditary diseases due to their lower immunogenicity and sustained gene expression [30]. Meanwhile, AdVs are widely applied in cancer immunotherapy and vaccine development, where their immunogenicity helps produce strong immune responses.

## 3. Evolution of Adenoviral Vectors for Gene Therapy

Three generations of adenoviral vectors represent the sequential development of vector systems aimed at eliminating the shortcomings of previous generations and improving their efficiency and safety (Figure 4).

The *E1A* gene (early gene 1A) and *E3* gene were first removed from AdVs in 1994 by Canadian scientist Graham and his research group, rendering the vectors replication-incompetent [31]. Although it has been shown that *E1*-depleted AdVs can recombine with the *E1* gene in 293 cells, whose genome contains Ad5 DNA sequences from nucleotide 1 to 4344 [32], resulting in the formation of replication-competent adenovirus (RCA). The contamination of AdVs with RCA is a significant problem for gene therapy. According to the Food and Drug Administration (FDA), the standard for RCA in the total amount of AdVs for clinical trials is less than 1 RCA per 3 × 10^10^ viral particles. A single patient can receive up to 10^13^ viral particles, so a small amount of RCA will still be present [33].

In 1998, Amalfitano and colleagues developed new, improved AdVs with deletions in *E1*, *E2b*, and *E3*, which have several enhancements, including a reduced risk of RCA formation and a larger insert size for the gene of interest. This made them potentially applicable for gene therapy [34]. In 2007, American scientists Campos and Barry modified E1A-depleted AdVs by removing the *E1B* and *E3* genes, reducing the RCA formation risk [35]. One of the advantages of first-generation AdVs is their large capacity, which allows them to include a significant amount of genetic information, making them useful for delivering large genes. Additionally, they are replication-incompetent vectors. Another advantage is the high level of transgene expression. However, the first-generation AdVs were highly immunogenic due to the presence of *E1* and *E3* genes, causing an immune response in the host, limiting their effectiveness and safety for clinical use. Nevertheless, the first-generation adenoviral vectors served as a starting point for developing more advanced variants, as all subsequent generations also lacked *E1* and *E3* genes. The most popular method for creating adenoviral vector constructs is the pAdEasy system developed by Bert Vogelstein [36].

To address this problem, all early gene sequences (*E1*–*E4*) were removed from the adenoviral vector, creating second-generation vectors. Second-generation adenoviral vectors were specifically designed considering several factors, such as reducing immunogenicity and RCA generation and increasing the cloned gene’s capacity. To achieve these goals, the second-generation AdVs had not only the *E1* and *E3* genes removed but also the non-structural *E2* and *E4* genes, increasing the cloning capacity to ~12 kb. Despite their improvements, second-generation AdVs still had limitations in transgene capacity and duration of expression.

Third-generation adenoviral vectors were created by removing most genes responsible for virus replication and replacing them with transgenes. These high-capacity adenoviral vectors, known as helper-dependent adenoviral vectors (HD-Ad) or HC-AdV (high-capacity adenoviral vectors) or gutless vectors, lack most viral genes and retain only ITRs and the packaging signal [37]. They can accommodate up to 37 kb of genetic information. Additionally, replicative-competent oncolytic adenoviral vectors capable of specifically replicating in tumor cells have been developed, making them promising for cancer treatment [38]. Third-generation adenoviral vectors are characterized by reduced immunogenicity and prolonged transgene expression, making them more promising for clinical use in gene therapy. However, the production of HC-AdV vectors is more complex as it requires a helper for replication and involves higher costs.

## 4. Clinical Trials Based on Adenoviral Vectors for Treating Hereditary Diseases

Since 1989, more than 4000 clinical trials for gene therapy have been conducted worldwide, with approximately 600 of these based on the use of adenoviral vectors; the studies were either initiated or completed [39]. However, in active trials, adeno-associated vectors are used more frequently than AdVs [40,41]. Despite their smaller packaging capacity, they have lower immunogenicity, which is a significant advantage over the highly immunogenic adenoviral vectors.

Analysis of the clinical trials database clinicaltrials.gov revealed that in active AdV trials for gene therapy, types such as Ad5 and Ad2 are most used. These types are the most extensively characterized. The genomes of Ad5 and Ad2 have been fully sequenced, showing approximately 95% similarity at the nucleotide level and a similar arrangement of transcription units [42]. Most clinical trials (around 70%) using adenoviral vectors are aimed at gene therapy for cancer [43]. Currently, there are already two methods of anti-tumor therapy based on AdVs, and some are in the late stages of clinical trials, indicating the success of these vectors for cancer gene therapy. This can be explained by the fact that the high immunogenicity of AdVs activates the host immune system, leading to a strong anti-tumor response that helps destroy tumor cells [44]. AdVs are also successfully used as platforms for vaccine development due to their high transduction efficiency and ability to elicit both innate and adaptive immune responses, which are crucial for effective immunization against infectious diseases. Adenoviral vector-based vaccines, such as those against Ebola and COVID-19, have demonstrated the ability to induce both humoral and cellular immune responses, leading to the production of neutralizing antibodies and T-cell responses that provide protection against target pathogens [45,46]. There are also emerging clinical trials of AdVs in the field of cardiovascular and degenerative diseases, but these are currently very few [47].

Analysis of the clinical trials database on clinicaltrials.gov revealed some clinical trials of adenoviral vectors for the gene therapy of hereditary diseases (Table 1).

### 4.1. Gene Therapy for Cystic Fibrosis

Adenoviral vectors have been used in gene therapy research for cystic fibrosis. Cystic fibrosis is a relatively common autosomal recessive disorder caused by mutations in the *CFTR* gene. The *CFTR* gene, 250 kb in length, includes 27 exons and is located on the long arm of chromosome 7 in region q31 [48]. The transmembrane protein encoded by the *CFTR* gene is an ion channel on the surface of epithelial cells involved in chloride ion transport. Mutations in the *CFTR* gene result in either the complete absence of the ion channel or impaired conductivity due to various protein changes. The complete absence of the CFTR protein due to nonsense mutations leads to the most severe form of cystic fibrosis [49]. Using third-generation adenoviral vectors with the *CFTR* transgene, Ad5-CB-CFTR (later renamed H5.020CBCFTR) and H5.001CBCFTR, two clinical trials (NCT00004779, NCT00004287) were conducted in which AdV was administered intranasally and via bronchoscopy into the respiratory epithelial cells, respectively [50,51].

The clinical trials and in vivo studies revealed several shortcomings of these vectors, limiting their use in further experiments. In situ hybridization of respiratory epithelial biopsies from trial participants in both cystic fibrosis gene therapy trials showed that only 1% of epithelial cells were transduced by the adenoviral vector, and potential measurements on the epithelial surface did not show significant restoration of Cl and Na transport. It is known that to restore CFTR channel function, correction of 5–10% of respiratory epithelial cells stably expressing the *CFTR* gene is required [52].

Preclinical trials of Ad-CFTR had previously been conducted in primates, with positive results forming the basis for continued research and leading to the approval of clinical trials in humans [53]. Adult male and female rhesus macaques (*Macaca mulatta*) weighing 3 to 10 kg were chosen for preclinical trials due to their physiological and anatomical similarities to humans, allowing for serial clinical studies with vector administration protocols like those in human trials. The doses of Ad-CFTR vectors ranged from 2 × 10^7^ to 1 × 10^11^ PFU for intrabronchial administration via bronchoscopy, and 1 × 10^11^ PFU for injection into the tracheal epithelium using an endotracheal tube.

Although the study did not provide quantitative data on transduction efficiency, it demonstrated successful gene transfer, as evidenced by *CFTR* gene expression in airway epithelial cells. Moreover, Ad-CFTR DNA was not detected in other organs. The restoration of CFTR channel function was not assessed in the study. Histological evaluation of lung sections after Ad-CFTR administration in macaques revealed side effects, namely dose-dependent lung inflammation. The inflammation was focal and characterized by the presence of perivascular macrophages and lymphocyte infiltrates. The degree of inflammation correlated with the administered vector dose, with higher doses leading to more pronounced inflammatory reactions. In the study, macaques receiving low doses of Ad-CFTR showed no noticeable differences in lung appearance compared to control animals. However, those receiving medium or high vector doses exhibited changes in a specific area of the right lower lung lobe. These changes included localized tissue density increase and a color change to reddish-brown, possibly indicating inflammation or hemorrhage. Despite the localized inflammation, clinical indicators such as body temperature, blood tests, serum chemistry, and leukocyte count, which are primary indicators of inflammation, remained within the control range, except for one animal that developed a fever. No systemic immune response was detected, as none of the animals receiving any dose of Ad-CFTR showed increased titers of neutralizing antibodies to adenovirus type 5 in their serum over the 21-day study period.

Thus, despite effective gene transfer leading to *CFTR* gene expression in primate respiratory epithelial cells, Ad-CFTR clinical trials in humans did not yield the expected results and had several issues. Firstly, the high immunogenicity of Ad-CFTR and the associated side effects of vector administration were problematic. In the first study, a patient in the high-dose group (2 × 10^10^ viral particles) showed an increase in neutralizing antibody titers from 1:80 to 1:1280 by 21 days post-vector administration. This could lead to enhanced immune responses upon vector re-administration. In the second trial, participants exhibited diverse cytokine secretion profiles of IL-2, IFN-g, IL-4, and IL-10. Most subjects had elevated levels of IFN-g and IL-10.

Each clinical trial identified side effects. In the first trial, participants receiving low vector doses (2 × 10^7^ viral particles) showed no significant changes in temperature, blood pressure, blood chemistry, or chest X-rays and had no local symptoms. However, two patients in the high-dose group experienced symptoms of toxicity within 12–24 h post-AdV administration. One patient had ear pain and tympanic membrane inflammation; another had jaw pain. Symptoms peaked at 48–96 h and fully subsided within three weeks.

In the second trial, subjects experienced flu-like symptoms, including weakness, nausea, muscle aches, and headaches. Additionally, one participant in the high-dose AdVs group (2.1 × 10^11^ viral particles) had a temperature above 38.3 °C, developed chest pain, and a focal infiltrate in the lung segment that persisted for more than four days. However, symptoms and infiltrates significantly regressed by day 10 [54].

Thus, using a higher vector dose was associated with inflammatory reactions in subjects, indicating that high vector doses can cause toxic effects, limiting their use. Increasing the vector dose did not affect gene therapy efficacy. The high immunogenicity of AdV-CFTR triggers cellular and humoral immune responses, manifesting as fever, temperature elevation, inflammation, and pneumonia. These side effects are associated with viral gene expression in cells. Moreover, the vectors did not provide sustained, stable transgene expression necessary for achieving therapeutic effects in treating hereditary diseases. It was found that repeated AdV dosing is unsafe and ineffective. Vector administration mediates immune responses and type-specific neutralizing antibody formations, reducing the effectiveness of repeated AdV administration. The same study showed that transgene expression upon the second administration was observed only in patients receiving a specific, but not the highest, vector dose, and was not observed at all upon the third administration [54]. Additionally, the authors note that no increase in neutralizing antibody levels was detected upon repeated AdV administration.

Furthermore, physical barriers to AdV penetration into target cells are significant. The human respiratory system is constantly exposed to the external environment, so numerous protective mechanisms exist against inflammatory reactions, mechanical damage, allergens, and pathogens. These mechanisms can be divided into two groups: physiological (immune) and physical. Innate and acquired immunity at the cellular and humoral levels provide airway protection. Physical barriers in the respiratory epithelium prevent adenoviral vectors from penetrating cells.

The mucus on the surface of the respiratory tract is a dense complex of mucoglycoproteins—gel-forming mucin fibers with negatively charged glycans and hydrophobic regions [55]. Some studies have demonstrated that this mucus is the greatest barrier to gene delivery [56,57,58], particularly for the penetration of adenoviral vectors [59]. Mucoglycoproteins have various bonds, such as ionic, hydrogen, hydrophobic, and electrostatic interactions, which trap inhaled foreign particles, including gene therapy vectors, in the mucus. These particles are then cleared from the lungs via mucociliary clearance, preventing the vectors from reaching target cells. Thus, the mucus layer of the respiratory epithelium, combined with mucociliary clearance, which protects the airways, is a significant barrier to vector penetration and transgene delivery, thereby reducing the effectiveness of gene therapy.

Studies have found that people with cystic fibrosis are characterized by the accumulation of mucus clumps adhered to the surface of the respiratory epithelium and impaired mucociliary clearance, providing favorable conditions for infections and inflammatory reactions [60,61]. Moreover, the average pore size of mucus in cystic fibrosis patients is much smaller than in healthy individuals, at 140 ± 50 nm [62]. This is because neutrophils release actin filaments and DNA, which bind and form polymers, increasing the viscosity of mucus in cystic fibrosis and enhancing its barrier properties [63,64]. The narrowing of mucus pore sizes can explain the inability of adenoviral vectors to effectively penetrate the mucus barrier to reach target cells and deliver transgenes due to the larger size of the vectors.

It has been identified that the CAR receptor is localized on the basolateral surface of columnar epithelial cells of the respiratory tract and is also part of epithelial tight junctions, mediating intercellular adhesion [65]. CAR interaction with adenoviruses is facilitated by its binding to the fiber knob domain, which determines the receptor’s ability to attach to viruses [9]. Thus, another reason for low transduction efficiency is the absence of the CAR receptor on the apical surface of respiratory epithelial cells. Equally significant barriers for AdVs are the epithelial tight junctions, which prevent the adenoviral vector from binding to CAR on the basolateral membrane and, consequently, from penetrating the cells, reducing the effectiveness of gene therapy for cystic fibrosis [66]. To achieve long-term therapeutic effects from gene therapy in the airways, stable transgene expression in self-renewing cells is necessary. In the airways, these cells are the basal cells of the lung, which are the primary type of stem cells attached to the basal lamina located at the base of the epithelial layer; thus, these cells do not have direct contact with the airway lumen [67]. While targeting basal cells is crucial for long-term stable transgene expression, AdVs’ access to these cells is limited.

Therefore, despite preclinical AdV-CFTR trials predicting the success of clinical trials due to *CFTR* gene expression in the respiratory epithelium of primates, only 1% of epithelial cells in the respiratory tract of humans were transduced, and this percentage did not increase with higher vector doses. Moreover, AdV-CFTR induced a strong immune response and side effects in humans at the highest vector dose, limiting its repeated administration.

### 4.2. Gene Therapy for Ornithine Transcarbamylase Deficiency

Ornithine transcarbamylase deficiency (OTCD) is an X-linked disorder characterized by a dysfunctional or deficient enzyme, ornithine transcarbamylase (OTC), which plays a crucial role in the urea cycle, converting ammonia into urea for excretion from the body. Due to this deficiency, ammonia and other toxic substances accumulate in the blood, leading to hyperammonemia.

A clinical trial of OTCD gene therapy using a second-generation adenoviral vector with the *OTC* transgene was conducted (NCT00004386). The vector was administered into the right hepatic artery of six groups of three to four people with increasing doses from group to group (2 × 10^9^–6 × 10^11^ viral particles/kg). The study revealed a low level of gene therapy efficiency—1% of AdV-transduced liver cells, as well as serious side effects, which led to the death of one of the subjects [68].

It is noteworthy that in preclinical trials of OTCD gene therapy conducted on mice, the adenoviral vector with the *OTC* transgene showed promising results. The studies demonstrated effective AdV transduction of hepatocytes and increased *OTC* transgene expression levels in the liver, leading to the correction of the metabolic defect following intravenous vector injection in adult mice with OTCD. Thus, mouse models showed that the adenoviral vector could potentially normalize metabolism by restoring enzyme activity in the liver to 80–90% compared to the control group of mice [69]. Preclinical trials of the adenoviral vector were also conducted on large mammals—primates. As a result of AdV administration, transgene expression was found in approximately 20–40% of the liver parenchyma in the target lobe, indicating a significant level of transduction efficiency. Transgene expression was observed in non-target lobes, but to a much lesser extent. The study also revealed transgene expression in the endothelial cells of the hepatic artery. However, by day 29, *OTC* expression in the baboon liver was almost absent, indicating a decline in transgene expression over time. Thus, the promising results of preclinical studies provide important data for the development and optimization of adenoviral vector-based gene therapy for application in clinical trials [70].

AdV-OTC is characterized by high immunogenicity, as it caused serious side effects in subjects. This trial led to the death of one patient, Jesse Gelsinger, who had partial OTCD, received the highest vector dose (6 × 10^11^ viral particles/kg). His immune response was characterized by high serum levels of interleukin-6 (IL-6) and interleukin-10 (IL-10), but normal levels of tumor necrosis factor-alpha (TNF-α) immediately after vector administration. Jesse’s immune system reacted sharply to AdV due to the presence of pre-existing antibodies and activated T-cells to the adenovirus type 5 capsid proteins. His condition progressed to systemic inflammatory response syndrome—a severe reaction characterized by widespread inflammation throughout the body, along with biochemical evidence of disseminated intravascular coagulation. As a result, Jesse died of multiple organ failure 98 h after vector administration. Postmortem studies revealed vector DNA sequences in most tissues, indicating widespread vector distribution and systemic immune activation [71].

Other subjects who received various vector doses (2 × 10^9^–6 × 10^11^ viral particles/kg) had peak IL-6 levels 8 h after AdV administration, especially in the high-dose vector groups (2 × 10^11^–6 × 10^11^ viral particles/kg), while IL-10 levels peaked 48 h after injection. Additionally, dose-dependent increases in IL-6 and IL-10 levels were observed, suggesting that higher AdV doses correlate with enhanced immune response/activation. Elevated levels of the cytokine TNF-α, which is involved in systemic inflammation, were not detected in any of the subjects.

Side effects manifested within the first 24–48 h after AdV administration. Participants experienced symptoms such as fever, muscle aches, nausea, and sometimes vomiting. The onset of fever usually occurred within 1–4 h after vector administration, with the temperature peak depending on the dose. Chills often preceded the temperature peak. A decrease in blood phosphate levels was also noted in some participants, but this was not clinically significant. Additionally, the high-dose vector group (6 × 10^10^ particles/kg) showed the most significant signs of biochemical liver damage due to increased aspartate aminotransferase levels in one participant, but levels returned to normal within 10 days.

Another issue with AdV-OTC is the availability of receptors for vector entry into cells. In situ hybridization showed transgene expression in liver biopsy samples from seven patients in no more than 1% of hepatocytes. There was no dose-dependent improvement in transduction efficiency. Metabolic indicators of urea synthesis and nitrogen accumulation were used to assess functional gene expression. A few patients with partial OTCD showed a slight increase in urea synthesis capacity and/or a decrease in orotic acid excretion in urine, but this was not statistically significant for the entire group of subjects [68].

The low efficiency of OTCD gene therapy can be explained by several reasons. Firstly, the authors of the clinical trial mentioned a study demonstrating methods of adenoviral vector administration and its biodistribution with each method [72]. Recombinant AdV was administered to mice and rats using various methods, including systemic intravenous injection into the tail vein, intracardiac injection into the left ventricular cavity, and microsurgical injection of the vector into the temporarily clamped aortic root. These model animals were chosen due to the conservative structure of the adenovirus binding domains in CAR.

Systemic intravenous administration via the tail vein primarily resulted in transgene expression in the liver. Variability in CAR expression between tissues and the lack of correlation between AdV expression patterns and viral receptor patterns suggest that anatomical barriers, particularly the endothelium, play a significant role in vector delivery and targeting specific cell types. Intracardiac injection aimed at achieving a higher local concentration of AdVs in the ascending aorta also resulted in high transgene expression levels in the liver: 91% in rat models. Microsurgical injection into the aortic root, bypassing hepatic filtration and improving delivery to the myocardium, significantly increased transgene expression in the heart. However, significant vector localization and notable transgene expression were also observed in the liver. This suggests that the liver remains the primary site of AdV transduction even with more targeted administration methods. Thus, in various administration methods in mice and rats, the adenoviral vector was predominantly found in the liver. Despite this, the OTCD gene therapy clinical trial revealed only 1% transduced liver cells.

In the same study, a comparative analysis of CAR mRNA expression in humans, mice, and rats was conducted [72]. It was found that human liver cells contain less CAR mRNA compared to mouse or rat liver cells. This means that human liver cells naturally produce fewer CAR proteins, which could explain why the adenoviral vector used in the OTCD gene therapy clinical trial had a low transduction level in humans. Moreover, the authors emphasize that despite the presence of CAR necessary for AdV entry into cells, there is no direct correlation between receptor expression levels and actual AdV targeting of specific cells. Therefore, the mere presence of receptors is not a prerequisite for successful gene transfer.

The study also mentions the vascular endothelium as an anatomical barrier that can impede adenoviral vectors from entering target cells, thereby reducing gene therapy efficacy. The liver is known to have endothelial pores up to 100 nm in diameter, a “wide-meshed” basal membrane, and sparse connective tissue. Thus, adenoviral vectors can penetrate cells of these organs if they have sufficient CAR protein expression levels. It can be concluded that the endothelium plays a crucial role as a barrier to AdV receptor access, consequently affecting vector transduction efficiency and transgene expression.

In summary, the transduction efficiency of adenoviral vectors is limited by their induced immune response due to the high immunogenicity of AdVs. Increasing the virus dose to enhance transduction can lead to various side effects, from fever and local inflammation to severe systemic inflammatory response syndrome and multiple organ failure, as evidenced by Jesse Gelsinger’s fatal outcome in the OTCD clinical trial. Adenoviral vectors exhibit broad tropism in various cell types due to widespread expression of the CAR in different tissues, facilitating virus binding and cell entry. This leads to non-specific AdV transduction of non-target cells or organs, and low CAR receptor expression by some cell types significantly reduces gene therapy efficiency and transgene expression. As shown, different methods of adenoviral vector administration do not lead to a significant increase in transgene expression in target organs. This can be explained by the variability in endothelial density and pore size, as well as differing levels of CAR expression by various human organs. Both factors together largely determine the efficiency of AdV target cell transduction.

The temporary nature of transgene expression with AdVs, due to their inability to integrate into the host genome, presents both challenges and advantages depending on the therapy. For gene replacement therapy, where long-term expression is crucial, this characteristic limits the use of AdVs. In such cases, integrating vectors are often preferred because they can provide long-lasting gene expression. However, using these vectors carries the risk of insertional mutagenesis, which can cause damage to the host genome and potentially lead to cancer [73].

On the other hand, AdVs remain as episomes in the host cell, which are suitable for genome editing and vaccine development, where temporary gene expression is sufficient. In genome editing, transient expression reduces the risk of off-target effects, making the therapy safer. Similarly, in vaccines, the temporary expression of antigens is enough to trigger a strong immune response, without the risks associated with integrating vectors [74].

## 5. Enhancing Adenoviral Vectors for Gene Therapy of Hereditary Diseases

### 5.1. Regulating Transgene Expression

Adenoviral vectors are highly effective in gene delivery; however, unregulated transgene expression can lead to cellular toxicity or immune-mediated clearance, which limits the therapeutic effect’s duration. To address these challenges, tissue-specific promoters are employed to restrict transgene activity to target cells. The main advantage of this approach is the reduction of expression in non-target cells to decrease drug toxicity, while the level of gene expression from such promoters is the same or even lower than when using universal promoters. For enhancing the efficacy of cystic fibrosis gene therapy, AdVs with promoters specific to airway cells have been developed, such as the 2 kb 5′-untranslated DNA sequences of the *CFTR* gene [75] and the k18 promoter—an epithelial cell expression cassette of keratin 18 [76]. Studies have shown a high level of reporter gene expression in airway epithelial cells of mice. These results indicate the potential of these vectors for use in cystic fibrosis gene therapy. The use of tissue-specific promoters has also been applied in OTCD gene therapy research. It has been shown that a liver-specific promoter influenced the production of neutralizing antibodies in mice, suggesting their use to circumvent the humoral immune response [77].

Although adenoviral vectors show great promise for advancements in gene therapy and have demonstrated success in vivo, the transition of these developments to clinical trials remains limited. In recent years, there has been a noticeable decline in research focusing on the design of AdVs with cell- or tissue-specific promoters for treating genetic disorders. Much of the foundational work in this area occurred during the 2000s, when gene therapy clinical trials with AdVs were more prevalent. This decline may be attributed to the fact that, beyond challenges in regulating transgene expression, adenoviral vectors face additional obstacles that restrict their widespread use in clinical applications for genetic therapies.

### 5.2. Capsid Alterations for Targeted Tropism Control

Adenoviral vectors often require elevated doses to achieve therapeutic gene expression, primarily due to their limited efficiency in transducing specific cell types. This inefficiency stems from the broad tropism of adenoviral vectors, which is linked to the widespread expression of CAR receptors across various cell types. Moreover, CAR is the primary receptor for adenoviral entry, but its presence on many different cells leads to low specificity. As a result, higher doses are necessary to ensure effective delivery to target cells, such as hepatocytes in OTCD therapy or respiratory epithelial cells in cystic fibrosis treatment. However, administering large vector doses can trigger dose-dependent toxicity and heightened immune responses, including the generation of neutralizing antibodies and cellular immune activity. These reactions can reduce gene transfer effectiveness and increase the risk of systemic inflammation, potentially causing severe adverse effects. Enhancing adenoviral specificity by refining their ability to target specific cell types has been a focal point in research, with capsid modifications emerging as a promising approach to address this challenge.

Capsid engineering has emerged as a promising solution to improve vector targeting. Techniques such as modifying capsid proteins or integrating receptor-specific ligands allow for precise delivery to target cells. For example, capsid pseudotyping—replacing capsid components with proteins from other viral types—has shown promise in retargeting vectors to receptors that are more accessible or specific to target tissues. In one study, the vector’s tropism was redirected from the CAR receptor, located on the basolateral membrane—difficult to access due to tight epithelial junctions between airway cells—to more accessible receptors on the apical surface of cells [78]. In another study, an Ad2(17f)/βGal-2 vector was developed based on adenovirus type 2 with Ad17 capsid proteins, which showed improved targeting of respiratory epithelial cells and increased transgene expression efficiency in vitro [79]. While pseudotyping enhances targeting, it may also increase off-target effects due to receptor overlaps on non-target cells. Addressing these limitations requires continued innovation in capsid engineering strategies, such as incorporating adapter molecules for receptor-specific binding.

Another effective approach to enhance adenoviral vector specificity involves the use of molecular adapters. These adapters, often bispecific molecules, bind simultaneously to vector capsids and target cell receptors, enabling selective targeting without the need for extensive genetic modifications. Adapter systems are used to remove the native tropism of the vector and create a new tropism towards the desired target cells [80]. A significant advantage of this method is that different adapters can bind to the same vector. To improve the effectiveness of gene therapy for cystic fibrosis, adapters have been employed as targeted ligands to enhance the specificity of AdVs toward respiratory epithelial cells. For instance, researchers identified the urokinase plasminogen activator receptor on the apical surface of the human airway epithelium and used a peptide adapter, connected to the adenoviral capsid via polyethylene glycol, to direct the vector to these cells. This approach significantly increased the transduction efficiency of epithelial cells in vitro [81]. An alternative approach of this method involves creating receptor-ligand complexes, where the natural receptor-binding ability of adenoviral capsids is replaced by ligands designed to interact with alternative receptors. For example, mutations in the adenovirus fiber gene, which normally interacts with CAR, redirected its binding preference to integrin, enabling gene transfer to cells that typically lack CAR expression. This adjustment in receptor specificity allowed the virus to successfully infect previously inaccessible cells in vivo [82]. Similarly, another study demonstrated that disabling the adenovirus’s ability to bind CAR effectively redirected its tropism to brain cells in vivo [83].

### 5.3. Overcoming Immune Barriers

Adenoviral vectors exhibit high immunogenicity, leading to elimination of transduced cells by the immune system, which shortens the duration of gene expression and increases the risk of immune responses and side effects. While single administrations are often sufficient for applications like genome editing, cancer therapies, or vaccines, treating hereditary diseases through gene replacement requires prolonged and stable transgene expression. Additionally, the presence of pre-existing antibodies against common adenovirus types limits the effectiveness of vectors derived from these strains. Furthermore, the strong immunogenicity of adenoviral proteins significantly restricts the frequency of their repeated use in the same patient. It is important to note that challenges related to pre-existing immunity and the limited efficacy of repeated administrations are not exclusive to adenoviral vectors; they are common across various viral gene therapy platforms. For instance, AAV vectors encounter similar obstacles due to pre-existing neutralizing antibodies against prevalent AAV serotypes, which can significantly diminish transduction efficiency in individuals previously exposed to these viruses [84].

Chemical modifications using polyethylene glycol (PEG), a hydrophilic and low-immunogenic polymer, enhance adenoviral vector properties by reducing immune responses and increasing transgene expression duration. The drawback of polymer attachment is the necessity to modify vector capsids each time after cell transduction and viral particle assembly, as changes occur not in the genetic information, meaning newly assembled vectors will not contain modifications [85]. Chemical modifications of protein drugs based on the covalent attachment of polyethylene glycol are often used in studies because it reduces their immunogenicity, increases solubility, and positively affects biological activity in vivo [86,87]. Preclinical studies on PEGylated adenoviral vectors demonstrated reduced immune activation, including lower cytotoxic T-lymphocyte activity and neutralizing antibody levels, alongside improved transgene expression in mouse models of cystic fibrosis [88,89].

Helper-dependent adenoviral vectors, lacking viral genes, enable long-term *CFTR* gene expression with minimal immune response, making them promising for cystic fibrosis therapy. Studies show that HD-Ad combined with CRISPR-Cas9 efficiently delivers the *CFTR* gene to respiratory epithelial cells [90]. Using cyclophosphamide for immunosuppression significantly reduced anti-HD-Ad antibodies and immune cell infiltration, allowing sustained *CFTR* expression with repeated delivery. This approach highlights the potential of HD-Ad with temporary immunosuppression for effective cystic fibrosis treatment [91].

### 5.4. Mucolytics for Breaking Airway Mucus Barriers

In diseases like cystic fibrosis, dense airway mucus poses a significant barrier to efficient gene delivery [56]. One of the most widely studied approaches in this category is the use of mucolytic agents that break down components of airway mucus. Temporarily disrupting epithelial tight junctions can enhance transduction efficiency and reduce the dose of the vector needed to achieve a therapeutic effect. Studies have demonstrated the application of various agents, such as L-α-lysophosphatidylcholine (LPC) [92], EGTA [93], polidocanol [94], sodium caprate [95], EDTA [96], calcium phosphate co-precipitation [97], and polycation [98]. Despite the increased transduction efficiency of respiratory epithelial cells and the enhanced expression level of the *CFTR* gene in vivo in animal models using mucolytics, these agents have not been used in clinical trials for cystic fibrosis gene therapy.

The likely reasons are that mucolytics do not achieve a consistent effect in increasing AdVs transduction efficiency. Additionally, the concentration and duration of administration need to be tailored for each individual patient. Furthermore, the introduction of some mucolytics along with the adenoviral vector can reduce its stability and functionality. Performing two separate administrations of the vector and the mucolytic significantly complicates the procedure for conducting trials.

## 6. Conclusions

The analysis conducted shows that the potential of adenoviral vectors for gene therapy is significantly limited. Due to the high immune response and transient nature of transduction, adenoviral vectors are not suitable for gene replacement therapy, especially when the target cells are dividing. In such cases, the effect will be short-lived, and repeated administrations of the vector are not feasible. Currently, the authors of this review believe that the primary applications for adenoviral vectors are in scenarios where temporary gene expression is required, such as in vector vaccines or genome editing.

Nonetheless, advancements in vector engineering, such as capsid modifications for targeted delivery and the incorporation of tissue-specific promoters, show promise in overcoming some of these limitations. Efforts to reduce immunogenicity through chemical modifications, including PEGylation, and the development of helper-dependent vectors with reduced viral elements offer potential to expand the utility of adenoviral vectors. Additionally, combining these technologies with mucolytic agents to address physical barriers, such as respiratory mucus, may further enhance their efficacy.

The authors of this review believe that modern advancements in controlling unwanted immune reactions may aid in developing new therapeutic directions using genome editing with adenoviral delivery. Ultimately, while current challenges restrict their use in long-term applications, continued innovation could unlock new possibilities for adenoviral vectors in treating hereditary diseases and beyond.

## Figures and Tables

**Figure 1 biology-13-01052-f001:**
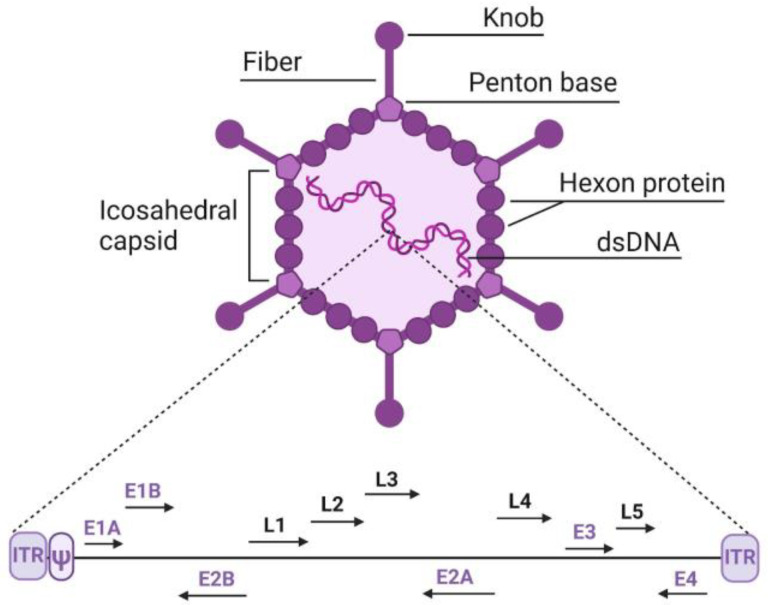
Structure of adenovirus and its genome. *E1*–*E4*—early genes, *L1*–*L5*—late genes. Explanation of other adenovirus elements is provided in the text.

**Figure 2 biology-13-01052-f002:**
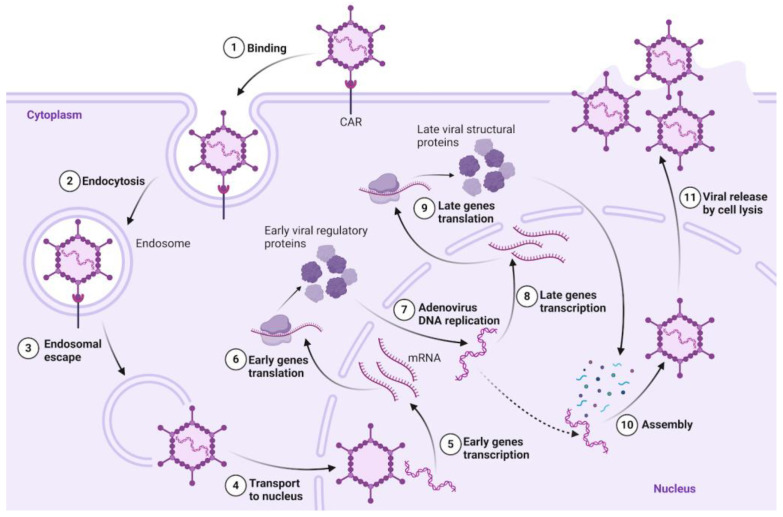
Adenovirus infectious cycle.

**Figure 3 biology-13-01052-f003:**
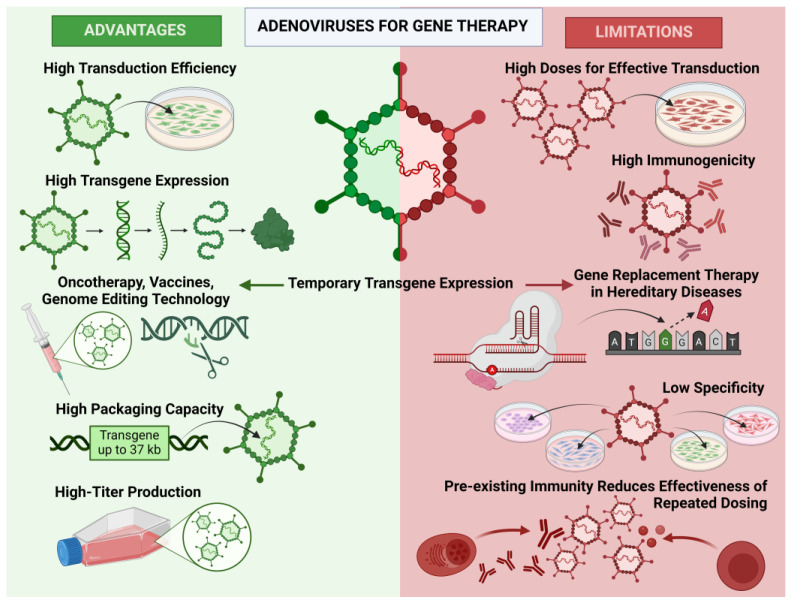
Advantages and limitations of adenoviral vectors in gene therapy.

**Figure 4 biology-13-01052-f004:**
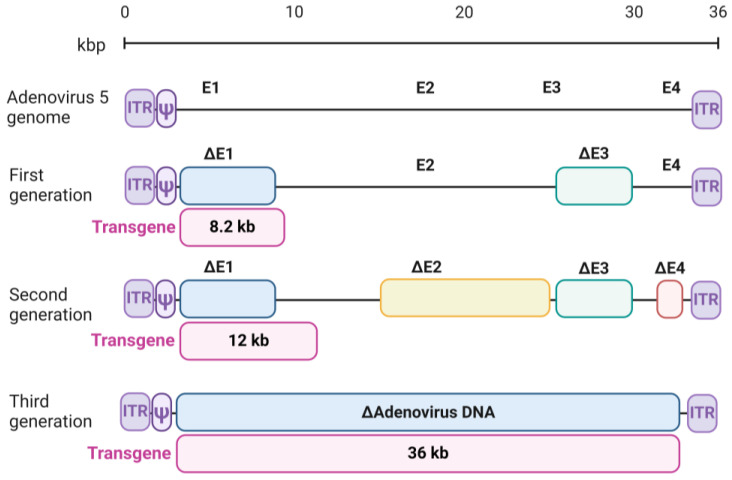
Generations of adenoviral vectors. *E1*–*E4*—early genes. The insertion sites for the transgene are shown below the deletion map.

**Table 1 biology-13-01052-t001:** Clinical trials of gene therapy for hereditary diseases with adenoviral vectors.

NCT Number	Study Title	Study Status	Vector	Phase	Enrollment	Study Years
**Cystic Fibrosis**						
NCT00004779	Phase I Pilot Study of Ad5-CB-CFTR, an Adenovirus Vector Containing the Cystic Fibrosis Transmembrane Conductance Regulator Gene, in Patients with Cystic Fibrosis	Completed	Ad5-CB-CFTR	Phase 1	12	1993–1995
NCT00004287	Phase I Study of the Third Generation Adenovirus H5.001CBCFTR in Patients with Cystic Fibrosis	Completed	H5.001CBCFTR	Phase 1	11	1999
**Ornithine Transcarbamylase Deficiency (OTCD)**						
NCT00004307	Study of Treatment and Metabolism in Patients with Urea Cycle Disorders	Unknown	AdV-OTC	Phase 1	66	1999
NCT00004386	Phase I Pilot Study of Liver-Directed Gene Therapy for Partial Ornithine Transcarbamylase Deficiency	Terminated	AdV-OTC	Phase 1	18	1999
NCT00004498	Phase I Study of Adenoviral Vector Mediated Gene Transfer for Ornithine Transcarbamylase in Adults with Partial Ornithine Transcarbamylase Deficiency	Terminated	AdV-OTC	Phase 1	18	1999

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
