# Peer review of "Adenoviral Vectors for Gene Therapy of Hereditary Diseases"

_biology, 2024, doi:10.3390/biology13121052_

Round 1
Reviewer 1 Report
Comments and Suggestions for Authors
The manuscript gives a good overview of some of the virotherapy applications of the HAdVs. As such, the manuscript is OK written, although better linguistics would make it even better to read. I should give a big credit o the authors for explaining in details some of the seminal studies where HAdV has been used for therapeutic purposes. Particularly the case with Jesse Gelsinger, as nicely explained by the authors. Also, the figures are aesthetically pleasing to look at, which attracts the reader to read the manus.
However, I do have some minor comments, which I hope that the authors will address in their revised version.
Here are my comments:
General comment: the authors should find a way how to write “adenoviruses”. Throughout the text, I could find it as hAdV-C5, AdV, adenovirus, adenovirus serotype 5 etc. Since the majority of the virotherapy studies deal with human adenoviruses (AdV), I strongly suggest the authors to define the acronyms (HAdV, AdV) at the early stage of the manuscript. Something like, “Human adenoviruses (hereafter as HAdV or AdV) ar known ……”).
L37&L44 & etc: we do not use serotypes, but rather types nowadays to describe HAdVs. Implement this change throughout the anus.
L44: there are more than 116 HAdV types. Use the Adenovirus Working group (http://hadvwg.gmu.edu/) as the reference.
L67: if the CAR is defined here no need to re-define it again (e.g., L78)
L83-85: “AdVs do not integrate into the host genome but remain inside the cell as an episome, providing temporary transgene expression”. This statement without a reference is too strong statement. To the best of my knowledge, there is not 100% evidence that hAdV can not be integrated into the genome. Even more contradicting: the HEK293 cell line contains HAdV genome (although parts of it), so it is possible that the HAdV can integrate into the genome. Update the sentence either by reference(s) or rephrase it.
L103: “What does the “Secondly” reflect to? Where is the first statement the authors point out?
L153-155: The statement, “One popular second-generation system is the pAdEasy system developed by Bert Vogelstein [24].” is not entirely correct as the AdEasy still has E4 and E2 regions left. From the author’s text, one feels that AdEasy will also lack the E2 and E4 regions, as it should be the second generation vector (also depicted in the figure). However, it is not true, or at least the Ref.24 in this context is incorrect. If one is very picky, the AdEasy vector by Vogelstein resembles more of the first-generation vector in Fig.3.
L306: No need for this statement again “It has been identified that the CAR receptor, which is an adenovirus receptor, i….”
L526 “ышву,” I guess it means “side” in English if I use my old Russian skills, correct?
Although figures 1-2 are clear and informative, they do not add much to the present review. I suggest the authors keep all the present figures and end the review with a new figure illustrating the pros and cons of using HAdV for virotherapy purposes. It can be a simple but informative drawing where the authors point out the pros and cons they have mentioned in the text. I have reviewed several HAdV and HAdV-based therapy reviews, but this kind of figure is missing in all of them. The authors are good at using Biorender, so why not make their own unique figure explaining the pros and cons? I am pretty sure this figure will be used by many adenovirologist 😊. An alternative would be to make a table with pros and cons, but that would not be as elegant and informative as the drawing.
Author Response
Dear Reviewer,
Thank you for your detailed analysis and constructive suggestions. We greatly appreciate the time and effort you put into reviewing our manuscript. All of your comments have been carefully considered and incorporated into the revised version, significantly improving the quality of the manuscript.
The changes are highlighted in the review mode, and we have provided responses to each of your comments in the comments section. Since you are Reviewer 1, please refer to the comments marked as "Reviewer 1" for specific responses.
Additionally, we found your suggestion to include a figure illustrating the pros and cons of AdV for gene therapy to be excellent, and we have added a new figure based on this idea.
Thank you once again for your valuable feedback.
Best regards,
Anna Muravyeva

Reviewer 2 Report
Comments and Suggestions for Authors
Muravyeva and Smirnikhina have generated a review of clinical trials utilizing adenoviral vectors for gene therapy of cystic fibrosis and ornithine transcarbamylase deficiency. The authors have presented important context to these trials by providing background about adenoviral vectors. Although the clinical trials presented in this review are not current, the topics reviewed remain important to the field. Viral gene transfer is a rapidly expanding field, with many viral gene therapy vectors in various stages of development and regulatory approval. The authors conclude that there is limited potential for adenoviral vectors for gene therapy based on clinical trial data from 25 years ago. While it is true that there are many weaknesses of adenoviral vectors as gene replacement therapy vectors, many strategies have been developed to combat these weaknesses since these trials were conducted. The authors have indicated that gene replacement therapy is transient, however helper-dependent adenoviruses have provided transgene expression in non-human primates for up to 7 years, and lifetime expression in mice. The large carrying capacity of the vectors also makes adenovirus a suitable method of delivering CRISPR-Cas9 gene editing technology. The authors should provide contemporary studies that have improved the described limitations adenoviral vectors, as much of the information provided is dated. As well, there are several points in the manuscript that need to be corrected to improve the quality of this review, as outlined below.
Major points:
The title of the review does not reflect the content. The review focuses only on clinical trials for cystic fibrosis and ornithine transcarbamylase deficiency, whereas the title suggests the manuscript will cover the innumerable preclinical studies that have explored adenoviral vectors for treatment of all hereditary disorders.
Throughout the manuscript, there are several paragraphs that have very few or no citations. In other cases, a review is inappropriately cited, and a study that experimentally supported the claim should be cited. Please ensure that all points within the manuscript are appropriately cited.
There are many instances where acronyms are defined multiple times, such as in the case of CAR. As well, there are other instances where an acronym is not defined prior to its use, and where an acronym is defined and then not used. Please correct all of the acronyms, including in the abstract where “adenoviral vectors” is written three times after “AdVs” has been defined.
An overview of what the various regions of the adenovirus genome do would inform the reader of what the effect of removing these regions is.
The authors highlight the unique benefits and strengths of adenoviral vectors, however these points could be emphasized if the authors provided context and compared adenovirus to adeno-associated virus or other commonly used gene transfer vectors.
Specific points:
Line 35: Please indicate if the 70-100 nm diameter is referring to the whole virion, including the capsid, fiber, and knob, or just the capsid.
Line 46: It should be mentioned that adenoviruses are found in many other species, and these viruses can be utilized for therapeutic applications, such as in the case of ChAdOX1.
Line 52: This sentence makes it seem as though adenovirus can only bind to CAR or CD46. Please correct this statement.
Line 64: It would be more accurate to indicate a specific region of the adenovirus genome is removed, which can be replaced by a transgene. The deleted region is not always replaced by a transgene, as is commonly the case with the E3 region.
Lines 93-118: This paragraph should be rewritten for clarity, as several points are repeated with slightly different wording.
Line 124 (Figure 3): This figure suggests that the transgene must always and only be contained within the E1 region, and transgenes cannot be placed within other regions, which is not the case, as transgenes can also be placed within the deleted E3 region, or even other regions of the genome. For example, first generation vectors are deleted of the E1 region, and frequently the E3 region, but the transgene can be cloned into a number of different regions of the vector.
Line 126: This statement is inaccurate. The cited article is an overview of adenoviral vector construction that was published by Drs. Graham and Prevec in 1995, however the initial removal of the E1 region was conducted much earlier… as described in the Graham and Prevec cited article.
Line 445: This statement makes it sound like vector integration is only beneficial. Insertional carcinogenesis, which is a real concern with integrating vectors, should also be discussed here.
Line 465: “in vivo” should be italicized, as it is throughout the rest of the manuscript.
Line 494: Please provide more information about these receptors.
Line 525: There is a word here in Cyrillic characters, please correct this.
Lines 528-531: It should be noted that issues with pre-existing antibodies and repeat administration is not unique to adenovirus, as this is also the case with most, if not all, other viral gene therapy vectors.
Lines 544-551: It should be indicated if this is a preclinical study.
Author Response
Dear Reviewer,
Thank you very much for your thorough and insightful review of our manuscript. We truly appreciate the time you dedicated to providing such detailed comments and suggestions. Your observations have been incredibly helpful, and we believe that the manuscript has significantly improved as a result of your feedback.
We have carefully addressed all your comments, and the revisions are clearly marked in the review mode. Additionally, we have included responses to each of your points in the comment section. Your observations were spot on, and we feel the manuscript is much stronger now thanks to your input.
The changes are highlighted in the review mode, and we have provided responses to each of your comments in the comments section. Since you are Reviewer 2, please refer to the comments marked as "Reviewer 2" for specific responses.
Once again, thank you for your valuable contribution to this work.
Best regards,
Anna Muravyeva
